# Skin-related adverse events and their associated factors among Diabetic patients on insulin therapy

Dennis Patson Mbwambo[1]*, Wigilya Mikomangwa[1], Rajabu Hussein Mkugwe[2], Bertha Mallya[1], Manase Kilonzi[1], Method Kazaura[3], Magreth Angelus[4], Kaushik Ramaiya[4], Mary Mayige[5], Ritah Mutagonda[1], Alphonce Ignace Marealle[1]

**1** Department of Clinical Pharmacy and Pharmacology, School of Pharmacy, Muhimbili University of Health and Allied Sciences, Dar es Salaam, Tanzania, **2** Department of Clinical Pharmacology, School of Biomedical Sciences, Muhimbili University of Health and Allied Sciences, Dar es Salaam, Tanzania, **3** Department of Epidemiology and Biostatistics, School of Public Health and Social Sciences, Muhimbili University of Health and Allied Sciences, Dar es Salaam, Tanzania, **4** Shree Hindu Mandal Hospital, Dar es Salaam, Tanzania, **5** National Institute for Medical Research, Tanzania

* dennis.mbwambo@gmail.com

## Abstract

### Background

More than 530 million individuals globally are afflicted with diabetes mellitus (DM), and the prevalence continues to escalate. Insulin remains the cornerstone of DM management across the globe. Nevertheless, existing literature indicates that individuals utilizing insulin are susceptible to developing abscesses and scar formation at injection sites. These complications may undermine therapeutic adherence, thereby adversely impacting the intended clinical outcomes. This study aimed to evaluate the prevalence of abscesses and scar formation at injection sites, along with their associated factors, among diabetic patients undergoing insulin therapy in Dar es Salaam, Tanzania.

### Methods

A hospital-based analytical cross-sectional study was conducted from the 28th of February 2024 to the 25th of May 2024. A total of 428 patients diagnosed with diabetes mellitus and undergoing insulin therapy were enrolled from four selected hospitals in Dar es Salaam. A validated case report form (CRF) was employed to gather socio-demographic characteristics and clinical data pertinent to the formation of abscesses and scars following insulin therapy. Data were analyzed utilizing Stata version 15.0 software, with findings summarized as frequencies and percentages. Factors associated with the development of abscesses and scarring were evaluated using modified Poisson regression, and a p-value of less than 0.05 was deemed statistically significant.

**Data availability statement:** All relevant data are within the manuscript.

**Funding:** The author(s) received no specific funding for this work.

**Competing interests:** The authors have declared that no competing interests exist.

## Results

Of 428 participants, the prevalence of abscesses and scar formation at the insulin injection site was 22.2% and 46.7%, respectively. Factors positively associated with abscesses were improper injection technique (adjusted Prevalence Ratio [aPR] = 1.11; 95% CI: 1.02–1.21, p = 0.009) and poor injection site rotation (aPR = 2.7; 95% CI: 1.13–6.45, p = 0.025). In contrast, the use of an insulin pen was negatively associated with abscesses (aPR = 0.13; 95% CI: 0.04–0.48, p = 0.002). Scar formation was positively associated with improper injection site rotation (aPR = 1.63; 95% CI: 1.03–2.32, p = 0.037) and uncontrolled blood glucose levels (aPR = 1.69; 95% CI: 1.01–2.84, p = 0.049).

## Conclusion

This study indicates that skin complications at insulin injection sites are highly prevalent. The findings suggest that improper injection technique, poor site rotation, and uncontrolled blood glucose are significant modifiable risk factors. The use of insulin pens may help reduce the risk of abscesses. Therefore, targeted patient education on correct injection practices and glycemic control is crucial to minimize these complications.

## Introduction

Globally, the prevalence and incidence of diabetes mellitus (DM) have escalated at an alarming rate [1]. In 2019, it was estimated that 530 million individuals worldwide were living with DM [2]. During the same period, the prevalence of DM among adults aged 20 years and older in Africa was recorded at 7.1% [3]. In West Africa and Asia, the prevalence of diabetes mellitus (DM) among the general populace is significantly elevated in urban areas compared to their rural counterparts; this disparity is estimated to exceed 10% [4]. In 2021, Tanzania was ranked among the top five nations in Africa with a notably high prevalence of DM, with an estimated 12% of the population affected by this condition [5]. The prevalence of diabetes mellitus (DM) within the general populace of Dar es Salaam has surged from less than 2% to nearly 10% within a mere two decades since 2001 [5,6].

Insulin therapy constitutes a fundamental component in the management of both type 1 and type 2 diabetes mellitus (DM) [7]. Notably, since 2017, we have observed a discernible correlation between the rising incidence of diabetes cases and the escalating consumption of insulin. In 2021, the global estimate of individuals with diabetes mellitus undergoing insulin therapy was approximately 100 million [8]. Based on the International Diabetes Federation (IDF) Atlas 11th edition 2023 report, the projected number of individuals utilizing insulin is anticipated to ascend to 192.9 million by the year 2030. Increasing trends in medicine consumption may be associated with adverse outcomes. For example, according to the FDA Adverse Events Reports, there was an escalating incidence of Diabetic Ketoacidosis (DKA) concomitant with

the heightened utilization of SGLT2 inhibitors from 2010 to 2022. Like wise, the burgeoning trend in insulin utilization may be associated with a myriad of safety concerns, particularly dermatological adverse events such as abscess formation and scarring at the injection sites. Abscesses and scars at the insulin injection site are cosmetically unacceptable and hinder optimal insulin absorption, thereby jeopardizing effective blood glucose regulation [9].The vigilant surveillance of insulin users is essential for the identification of previously unrecognized adverse events. The prompt identification and documentation of adverse events are imperative to mitigate their incidence and avert widespread detriment [10].

The discernible trend of escalating insulin utilization on a global scale, particularly within low- and middle-income countries (LMICs) such as Tanzania, necessitates a comprehensive evaluation of adverse events among insulin users. This scrutiny focuses specifically on the prevalence of abscesses and scar formation at the insulin injection site, along with the associated contributing factors.

## Materials and methods

### Study design and settings

A hospital-based analytical cross-sectional study was conducted from 28th February 2024–25th May 2024 in selected hospitals within Dar es Salaam. As Tanzania's largest and most densely populated city, Dar es Salaam is home to approximately 7.8 million inhabitants, constituting 12% of the nation's total population. Consequently, the selected hospitals epitomize the broader healthcare landscape of the region. Furthermore, existing literature indicates that the prevalence of diabetes mellitus among the general population in Dar es Salaam has escalated significantly in comparison to other regions of Tanzania.

From a selection of 10 public and 10 private hospitals in Dar es Salaam, each equipped with diabetic clinics and a substantial patient population suffering from diabetes mellitus (DM), four hospitals were randomly chosen: two public institutions (Mwananyamala and Temeke Regional Referral Hospitals) and two private establishments (Shree Hindu Mandal and Hubert Kairuki Memorial Hospital). These four hospitals are categorized as secondary-level facilities within the Tanzanian healthcare system. Each institution is staffed with healthcare professionals specializing in diabetes management. Collectively, these facilities accommodate a considerable influx of DM patients on a monthly basis; specifically, Shree Hindu Mandal serves at least 1,007 patients, Mwananyamala Regional Referral Hospital caters to 670 patients, Temeke Regional Referral Hospital treats 560 patients, and Hubert Kairuki Memorial Hospital manages 502 patients.

### Study population

This study encompassed patients diagnosed with diabetes mellitus (both Type I and Type II) who were undergoing regular insulin therapy and attended clinics at four designated healthcare facilities. Only those patients who had been utilizing insulin for a minimum of four weeks prior to the commencement of data collection were included in the study. The primary rationale for this timeframe was that a duration of four weeks could be adequate for any abscesses or scars to manifest on the skin. Additionally, this period allows for a thorough evaluation of the patients' practices regarding self-administration of insulin. For instance, the abdominal region should ideally be segmented into four quadrants, with each quadrant designated for use on a weekly basis. Consequently, a minimum duration of four weeks is plausible for assessing patients' injection practices, particularly with regard to site rotation.

Patients with pre-existing abscesses or scars at the insulin injection sites prior to the commencement of insulin therapy were excluded from the study. This exclusion was implemented to prevent an overestimation of the prevalence of adverse events, as it is plausible that some scars may have existed prior to the initiation of insulin treatment.

### Sample size and sampling technique

A total of 428 DM patients on insulin therapy were enrolled in this study. The formula for a cross-sectional study ($n = p(1-p) z^2/d^2$) was used to calculate the sample size [11]. Due to a lack of similar studies in Tanzania, we assumed a prevalence

of 50%, a confidence level of 95% (z = 1.96), and precision (d) of 5%. A minimum of 385 participants was obtained and assuming a 10% non-respondent rate, a total of 428 participants was obtained.

Three-stage sampling methodology was employed to procure the study participants. Initially, we compiled a list of the top ten [10] public healthcare facilities that catered to a substantial number of diabetes mellitus (DM) patients over the preceding three months. Subsequently, we randomly selected two [2] public healthcare facilities from this distinguished list utilizing a ballot box method. A parallel procedure was enacted to identify two [2] private healthcare facilities from the roster of the top ten [10] private hospitals. In the final stage, a systematic sampling was implemented with a defined interval (k = 2), the first patient was chosen at random in each clinic session until the requisite sample size for each facility was attained. The number of study participants contributed by each facility was determined by the ratio of patients served by that facility relative to the total DM patients treated across all four facilities.

## Data collection tools

A case report form (CRF) was meticulously developed through an extensive literature review, consultations with subject matter experts, and the investigators' substantial experience in conducting quantitative research. The CRF served as a pivotal instrument for the collection of data from both patient records and direct patient interviews. The tool underwent a pretesting phase involving 30 participants at Amana Regional Referral Hospital (RRH), a secondary healthcare facility in Dar es Salaam that, while not one of the designated study sites, shares analogous settings with the primary research locations. This pretest facilitated the refinement and modification of the tool prior to the commencement of data collection.

Four research assistants, one assigned to each site, were meticulously recruited and trained to gather data utilizing the CRF. These research assistants possessed degrees or diplomas in medicine or nursing and were actively engaged in the diabetic clinics at their respective study sites. To augment their comprehension of the study and to familiarize them with the tool, the research assistants participated in the pretesting phase.

## Definition and measurement of key variables

The occurrence of abscesses and scars at the insulin injection site were the key outcome variables. The variables examined for their association with the abscesses and scars formation at insulin injection site have been described (Table 1).

## Data collection procedure

Before enrollment, eligible patients were sensitized about the study. Those who showed interest were given the consent form to read and understand. Both English and Swahili versions were used at the subject's convenience. Those incapables of reading were supported by their guardian or the study team. Following an understanding of the study, patients who were willing to join the study were asked to sign or thumb the consent form.

Data were meticulously gathered through a structured process during a single study visit, comprising three key components: a patient interview, a physical examination, and the extraction of baseline clinical data.

A structured interview was administered to each participant to collect sociodemographic information and assess injection practices. Patients were asked to enumerate all anatomical regions recommended for insulin injection; those who successfully identified the abdomen, thigh, upper arm, and buttocks were classified as "aware," while others were classified as "unaware." Injection site rotation was assessed by querying patients on their practices. Correct technique was defined as systematically dividing the abdomen into four quadrants, using each for one week, and other sites into two halves, also used weekly, while ensuring consecutive injections were at least 1 cm apart. Furthermore, patients were specifically asked about the development of any abscesses or scars within the twelve months preceding the interview.

Concurrently, a focused physical examination of potential insulin injection sites was conducted by a trained study clinician in a private room to ensure patient comfort and confidentiality. This examination aimed to identify the objective

**Table 1. Definition and measurement of key variables.**

| Variable | Definition/Measurement |
|---|---|
| Abscess | A localized collection of pus that forms within tissues at the location where insulin was administered. |
| Scar | A localized tissue damage resulting from repeated insulin injections |
| Aware of insulin injection sites | Ability to enumerate all recommended sites for insulin administration. These include the lower abdomen, thigh, upper arms, and buttocks. |
| Good injecting practice | Ability to demonstrate the correct technique for insulin injection. This encompasses the appropriate angling of the needle, contingent upon whether the patient is lean or overweight. |
| Proper injection site rotation | A proficient ability to partition the injection area into quadrants and utilize each quadrant on a weekly basis, while ensuring a spacing of 1 cm between subsequent injections within a given quadrant. |
| Hygiene practice at injection site | Hygiene was considered good if patient cleaned the injection site before elf insulin administration. The site can be cleaned with water -soaked cotton balls or alcohol swabs before each injection. Otherwise it was considered ad poor hygiene practice. |
| Reuse of needle | Using the same needle to administer insulin on multiple occasions. |
| The source of insulin and syringes | This pertains to whether insulin and syringes were procured from the healthcare facility where the patient attends the diabetic clinic or from an alternative source. |
| Controlled average blood glucose | At least three random blood glucose readings recorded during the past three consecutive clinic visits should fall within the normal range (4.4–10.0 mmol/L). Otherwise, it is deemed uncontrolled.<br>Nb: HbA1c, a conventional metric for assessing long-term glycemic control, was not utilized due to the lack of affordability and accessibility of this test in those facilities.<br>RBG was preferred over Fasting Blood Glucose because RBG readings provides a more holistic "snapshot" of a patient's daily glucose fluctuations and overall exposure to hyperglycemia, which is a crucial aspect of diabetes management. |
| Insulin storage | The storage conditions were deemed appropriate if the insulins were stored within the acceptable range of 2–8 degrees Celsius. |

presence of any existing abscesses, scars, or other skin reactions, and its findings constituted the primary outcome measures for the study.

Additionally, to characterize the study population and assess their current level of glycemic control, the most recent random blood glucose (RBG) readings documented in the patient's clinic file over the preceding three months were extracted. This data was used solely to calculate an estimated mean blood glucose level for each participant at the time of the study visit, serving as an indicator of whether their glycemic control was adequate or not. No longitudinal tracking of glucose values was performed.

## Data analysis

Data were entered into MS Excel and transferred to Stata version 15.0 for analysis. Continuous variables were categorized based on clinical and analytical rationale. The age of participants was grouped into three life-stage categories (0–24, 25–44, and ≥45 years) to reflect distinct periods of potential behavioral, social, and physical capacity that could influence insulin injection practices and the risk of adverse events. This categorization also facilitated a more interpretable analysis

Frequency and percentages were used to summarize the findings. Factors associated with abscess and scar formation were assessed systematically using the chi-square test. Associations between predictors (needle length, comorbidities, insulin type, injection technique, etc.) and the outcome of interests (abscess and scaring), were assessed using multivariable Poisson regression. Poisson regression was chosen because the high prevalence of the outcomes (>10%) means that Odds Ratios from logistic regression would not accurately approximate the Risk Ratios and could overestimate the strength of associations. The models adjusted for demographics and clinical confounders determined a priori. Variables with a p-value<0.05 in the final model were considered factors associated with the formation of abscesses or scars at the injection site. Post hoc power was estimated using methods for multi-variable Poisson regression [12].

## Ethics statement

The study protocol was reviewed and approved by the Muhimbili University of Health and Allied Sciences (MUHAS)-Research and Ethics Committee (MUHAS-REC) with a reference number DA282/298/01.C/2057. The official permission to collect data was issued by the medical officer in charge of the respective health facilities. Each participant gave informed consent before data collection, and ethics were observed through data collection as stipulated in the Declaration of Helsinki.

## Results

### Description of the study participants

A total of 428 diabetic patients were enrolled in the study. The mean age of the participants was 38.4 years (SD ± 19.4), with a significant proportion being adults aged ≥45 years (44.6%). Females constituted 54.4% (233) of the participants, while 59.6% (255) were unemployed, and 70.3% (301) possessed health insurance coverage. Furthermore, a mere 1.9% of the participants identified as smokers, and 6.3% acknowledged alcohol consumption (Table 2).

### Patient awareness of insulin injection procedures and their injecting practices

Twenty-three percent (102/428) of the participants reported the presence of comorbidities other than diabetes mellitus (DM). Among these individuals, 11% (47) indicated a history of skin allergies. The predominant syringe employed for insulin administration featured a needle length of ≤ 6 mm, representing 55.6% of the sample. A significant majority, 93% (398/428), acknowledged reusing syringe needles, with the frequency of reuse ranging from a single injection to more than three. Furthermore, 88% (378/428) of the participants procured their insulin and syringes from the health facilities associated with their clinic visits. Notably, 79.4% (339/428) adhered to the stipulated conditions for insulin storage. At least 65% (278/428) of the study participants were aware of the anatomical sites recommended for insulin injections to ensure optimal absorption, and an impressive 90.2% (386/428) demonstrated proficiency in correct insulin administration techniques. However, a concerning 63.2% (273/428) of participants were unaware of the necessity for injection site rotation.

Table 2. Sociodemographic and clinical characteristics of study participants.

| Variables | Category | Frequency | Percent |
|---|---|---|---|
| Age (years) | ≤ 24 | 141 | 33.0 |
| | 25-44 | 96 | 22.4 |
| | ≥ 45 | 191 | 44.6 |
| Sex | Male | 195 | 45.6 |
| | Female | 233 | 54.4 |
| Level of education | Non-primary education | 185 | 43.2 |
| | Secondary education | 182 | 42.5 |
| | University | 61 | 14.3 |
| Patient occupation | Unemployed | 255 | 59.6 |
| | Self-employed | 109 | 25.5 |
| | Employed | 64 | 14.9 |
| Health insurance | Have health insurance | 301 | 70.3 |
| | No health insurance | 127 | 29.7 |
| Cigarette smoking | Yes | 8 | 1.9 |
| | No | 420 | 98.1 |
| Alcohol consumption | Yes | 27 | 6.3 |
| | No | 401 | 93.7 |

Additionally, only 32.2% (138/428) consistently sanitized the injection site prior to administering insulin. The investigation further revealed that 79.4% (340/428) of participants exhibited poorly controlled average blood glucose levels (Table 3).

### Prevalence of abscesses and scars at the insulin injection sites among DM patients

Of the 428 study participants, the prevalence of abscess formation was 95(22.2%) and 200(46.7%) for scar occurrence. Most 52(54.7%) of the observed abscesses are mild and the majority, 207(48.4%), occur following the use of mixed/fused insulin (Table 4).

### Factors associated with abscess formation at the insulin injection site among DM patients

In the adjusted analysis, improper injection technique (aPR = 1.11; 95%CI: 1.02–1.21; p = 0.009), being self-employed (aPR = 2.34; 95%CI: 1.04–5.27; p = 0.04), and inadequate injection site rotation (aPR = 2.7; 95%CI: 1.13–6.45; p = 0.025)

**Table 3. Patient awareness of insulin injection procedures and their injecting practices.**

| Variables | Category | Frequency | Percent |
|---|---|---|---|
| The source of insulin and syringes | At the health facility | 378 | 88.3 |
| | Other Places | 50 | 11.7 |
| Type/insulin formulation | Single separate formulations | 95 | 22.2 |
| | Mixed/fused Formulation | 207 | 48.4 |
| | Insulin Pen | 126 | 29.4 |
| Needle length in 'mm'' (n = 378) | ≤ 6 mm | 238 | 55.6 |
| | Insulin pen needle | 125 | 29.2 |
| | Unknown | 65 | 15.2 |
| Ever reused a syringe needle | Yes | 398 | 93 |
| | No | 30 | 7 |
| How often reused the needle (n = 398) | Sometimes | 134 | 31.3 |
| | Often | 137 | 32 |
| | Always | 127 | 29.7 |
| Frequency of needle re-use (n = 398) | Up to three shorts | 217 | 57.7 |
| | More than three shorts | 181 | 42.3 |
| Aware of injection sites | Know all the sites | 278 | 65 |
| | Know a few of the sites | 150 | 35 |
| Insulin injecting technique | Correct | 386 | 90.2 |
| | Not correct | 42 | 9.8 |
| Injection site rotation | Rotates correctly | 155 | 36.2 |
| | Does not rotate correctly | 273 | 63.8 |
| Clean injection site | Sometimes | 290 | 67.8 |
| | Always | 138 | 32.2 |
| Cleaning procedure (n = 253) | Correct | 165 | 65.2 |
| | Not correct | 88 | 34.8 |
| Average blood glucose | Well-controlled | 88 | 20.6 |
| | Not controlled | 340 | 79.4 |
| Insulin storage | 2 to 8ºC | 339 | 79.4 |
| | >8ºC | 89 | 20.6 |
| Presence of any comorbidity | Yes | 102 | 23.8 |
| | No | 326 | 76.2 |

**Table 4. Prevalence of abscess and scar at injection site among DM patients on insulin therapy (n = 428).**

| Variables | Category | Frequency | Percent |
|---|---|---|---|
| | | | |
| Skin allergy | Yes | 47 | 11.0 |
| | No | 381 | 89 |
| Ever developed an abscess | Yes | 95 | 22.2 |
| | No | 333 | 77.8 |
| When abscesses happened (n = 95) | Between Jan-Mar 2023 | 33 | 34.7 |
| | Between April and June 2023 | 12 | 12.6 |
| | Between July-Sept 2023 | 16 | 16.8 |
| | Between Oct and Dec 2023 | 34 | 35.9 |
| Seriousness of the abscess (n = 95) | Mild resolved within a few days | 52 | 54.7 |
| | Needed Medication | 36 | 37.9 |
| | Severe Needed Hospitalization | 7 | 7.4 |
| Ever developed a scar at the injection site | Yes | 200 | 46.7 |
| | No | 228 | 53.3 |
| When scars happened (n = 200) | Between Jan-Mar 2023 | 45 | 22.5 |
| | Between April and June 2023 | 23 | 11.5 |
| | Between July-Sept 2023 | 24 | 12 |
| | Between Oct and Dec 2023 | 108 | 54 |

were associated with a higher prevalence of abscess formation. The use of an insulin pen was associated with a lower prevalence compared to the use of vials and syringes (aPR = 0.13; 95%CI: 0.04–0.48; p = 0.002) (Table 5).

### Factors associated with scar formation at the insulin injection site among DM patients

In the adjusted analysis, inadequate injection site rotation (aPR = 1.63; 95% CI: 1.03–2.32; p = 0.037), uncontrolled blood glucose levels (aPR = 1.69; 95% CI: 1.01–2.84, p = 0.049), and being employed (aPR = 2.0; 95% CI: 1.21–3.33, p = 0.007) were associated with a higher prevalence of scar formation (Table 6).

### Discussion

The present study evaluated the prevalence of abscesses and scar formation at insulin injection sites, as well as the associated factors among patients with diabetes mellitus attending clinics in selected private and public hospitals in Dar es Salaam, Tanzania. The prevalence of abscesses, recorded at 22.2%, is notably higher than the figures reported from the United States and England [13,14]. According to these studies, the prevalence of abscesses has been reported as infrequent. One potential explanation for the observed variability in prevalence may stem from differences in study location, demographic composition, and healthcare systems. Additionally, the research conducted in England focused exclusively on pediatric populations, whereas this investigation encompassed individuals across all age groups. The findings from the United States, as reported by Richardson, were derived from a study conducted several years prior; this temporal aspect could further elucidate the discrepancies in the prevalence of abscesses. Moreover, the rising incidence of this adverse event may correlate with the increasing number of insulin users, as insulin is presently utilized by both type 1 and type 2 diabetes mellitus patients [8].

The elevated prevalence of scars (46.7%) observed in this study is consistent with findings from the USA [14], where Richardson et al. reported a prevalence of 48%. However, the actual burden reported by this study might have been underestimated. This is because during the recruitment of study participants, patients who had preexisting abscesses or

**Table 5. Factors associated with abscess formation at the insulin injection site.**

| Variable | N | Abscess formation (%) | Unadjusted | | Adjusted | |
|---|---|---|---|---|---|---|
| | | | cPR,95% CI | p-value | aPR,95% CI | p-value |
| **Age group** | | | | | | |
| ≤24 | 141 | 42 (29.79) | Ref | | | |
| 25-44 | 96 | 24 (25.00) | 0.83(0.96-1.09) | 0.41 | 0.99 (0.65-1.53) | 0.99 |
| ≥45 | 191 | 29 (15.18) | 0.51(1.03-1.14) | 0.002 | 1.06 (0.99-1.15) | 0.06 |
| **Sex** | | | | | | |
| male | 195 | 45 (23.08) | Ref | | | |
| female | 233 | 50 (21.46) | 0.93(0.97-1.06) | 0.689 | 1.02(0.97-1.08) | 0.87 |
| **Occupation** | | | | | | |
| Non | 255 | 51 (20.00) | Ref | | | |
| Self-employed | 109 | 30 (27.5) | 1.38(0.93-2.04) | 0.290 | 2.34 (1.04-5.27) | 0.04 |
| Employed | 64 | 14 (21.88) | 1.09(0.64-1.84) | 0.572 | 1.14 (0.39-3.32) | 0.815 |
| **Source of insulin and syringes** | | | | | | |
| Health facility | 378 | 79 (20.90) | 0.94(0.86-1.02) | 0.119 | 0.91 (0.79-1.03) | 0.13 |
| Other sources | 50 | 16 (32.00) | Ref | | | |
| **Site rotation** | | | | | | |
| Yes | 155 | 28 (18.06) | Ref | | | |
| No | 273 | 67 (24.54) | 1.36(0.91-2.01) | 0.108 | 2.7 (1.13-6.45) | 0.025 |
| **Insulin formulation** | | | | | | |
| Single vial | 95 | 18 (18.95) | Ref | | | |
| Mixed/fused | 207 | 66 (31.88) | 1.68(1.06-2.67) | 0.034 | 1.45 (0.59-3.53) | 0.42 |
| Insulin Pen | 126 | 11 (8.73) | 0.19(0.13-0.93) | 0.012 | 0.13 (0.04-0.48) | 0.002 |
| **Needle length** | | | | | | |
| ≤ 6mm | 238 | 69 (28.99) | 1.03 (1.01-1.05) | 0.32 | 0.94 (0.83-1.08) | 0.43 |
| Unknown | 65 | 15 (23.08%) | 0.91 (0.85-0.98) | <0.01 | 1.12 (0.97-1.29) | 0.11 |
| Pen needle | 125 | 11 (8.80) | Ref | | | |
| **Injecting technique** | | | | | | |
| Correct | 386 | 178 (46.11) | Ref | | | |
| Not correct | 42 | 22 (52.38) | 1.11 (1.07-1.16) | <0.01 | 1.11 (1.02-1.21) | 0.009 |

*The model was adjusted for blood glucose level, insulin storage practices, comorbidity, source of insulin and syringes, and demographic information (age, sex, occupation).*

scars at insulin injection sites prior to initiated of insulin therapy were excluded from the study. Therefore, patients with preexisted abscesses or scars who also possibly developed abscesses or scars after the initiation of insulin therapy were not captured. This implies that the problem is significant and exists globally irrespective of geographical location, economic status, or ethnicity.

Therefore, conducting pharmacovigilance for patients engaged in prolonged medication regimens, including those utilizing insulin, is of utmost importance. Pharmacovigilance will facilitate the prompt identification of adverse events, thereby mitigating potential harm to a significant segment of the population [15]. However, the surveillance of drug safety and the rates of reporting are markedly deficient in Tanzania [16]. Inadequate reporting can obscure the severity of the prevailing issue, permitting it to impact a substantial segment of the population.

We noted that the neglect of appropriate injection site rotation was associated with a 2.7 and 1.63-fold increase in the prevalence of abscesses and scars, respectively. This finding corresponds to what was observed in Ethiopia [17]

and India [18]. Similar to the findings elucidated in this study, Negashi et al. and Barua et al. documented a commendable practice of rotation at injection sites among patients with diabetes mellitus. This may indicate the similarities in the practices of insulin users across sub-Saharan Africa and Asia. Ignoring the crucial requirement of practicing injection site rotation results in trauma caused by consecutive injections at a single dermal site, ultimately leading to the emergence of abscesses [17,19].

A favorable association between the occurrence of abscesses and inadequate insulin injection methodologies, especially the injection angle noted in our study (indicating a 2.7-fold escalation in prevalence), reinforces the observations recorded in Bangladesh [9] and India [18]. In spite of the variations in study design, participant characteristics, and geographical settings, this analysis, in conjunction with the prior two studies, has highlighted a noteworthy positive association between the presence of abscesses and the inadequate execution of insulin injection techniques. It is recommended that the syringe be positioned at an appropriate angle, based on the particular thickness of the patient's skin [20]. Divergence from the prescribed injection techniques could be linked to the development of abscesses at the injection site.

Occupational activities were positively correlated with the incidence of abscesses, exhibiting an approximate twofold increase in prevalence for both formal and informal employment. Nonetheless, we were unable to identify a convincing scientific explanation to validate the noted association.

In relation to other studies, Esmail et al. indicated that certain occupational activities are linked to dermatological conditions [21]. However, Esmail did not probe further into the abscess or the scars, nor did our study broaden its inquiry into the variety of occupational activities. Consequently, this correlation remains unresolved until additional research is undertaken to provide definitive information.

An eighty-seven percent reduction in the prevalence of abscesses at insulin injection sites among diabetic patients utilizing insulin pens, as opposed to those employing conventional syringes and needles, suggests that use of insulin pens is negatively associated with abscesses formation at injection sites. This is not a novel situation, as comparable findings have been noted in Saudi Arabia, and Lebanon [22]. Studies indicate that insulin pens facilitate seamless insertion, minimize discomfort, and reduce dermal trauma; thus, they emerge as a preferred option among patients with diabetes mellitus who are engaged in regular insulin therapy [23].

Inadequate control and supervision of blood glucose levels within the recommended range demonstrated a positive association with the development of scar tissue at the insulin injection sites among patients with diabetes mellitus. One plausible explanation is that chronic hyperglycemia accelerates the formation of Advanced Glycation End Products (AGEs) [24]. AGEs bind to collagen in the skin and subcutaneous tissue, resulting in diminished elasticity and aberrant cross-linking that stiffens the skin, rendering it more susceptible to scarring. Another rationale is that hyperglycemia compromises microvascular integrity, thereby reducing blood flow to injection sites; as a consequence, microtrauma from injections heals at a slower rate, often leading to fibrotic scarring. Studies indicate that patients with an HbA1c level exceeding 8% exhibit 2 to 3 times higher rates of injection site complications [25]. It is essential to recognize that the presence of a scar at the injection site can profoundly impact the absorption of insulin, thereby influencing glycemic regulation. However, due to the nature of the study, establishing a temporal relationship between the abscess and dysregulated blood glucose levels proved to be challenging.

## Limitations

We adjusted for demographic and clinical covariates, however, residual confounding by unmeasured factors (e.g., genetic predisposition to scarring, precise skin thickness measurements, detailed lifetime injection history, overall diabetes self-care adherence) remains possible. Measurement issues (particularly regarding glycemic control) as we used the random blood glucose tests instead of the Hb1c. The cross-sectional design has some recall bias and limits causal inference.

This study's statistical power was a function of the number of cases for each outcome. For scar formation (200 cases), the study had > 99% power to detect a prevalence ratio (PR) of 1.5, but for abscesses (94 cases), the power was 79% for

**Table 6. Factors associated with scar formation at the insulin injection site.**

| Variable | Total | Developed scar (%) | Crude estimates | | Adjusted estimates | |
|---|---|---|---|---|---|---|
| | | | cPR,95% CI | p-value | aPR,95% CI | P-Value |
| **Age** | | | | | | |
| ≤24 | 141 | 68 (48.23) | Ref | | | |
| 25-44 | 96 | 50 (52.08) | 1.07 (0.89-1.06) | 0.561 | 0.96 (0.88-1.05) | 0.44 |
| ≥45 | 191 | 82 (42.93) | 0.89 (0.96-1.11) | 0.340 | 1.02 (0.95-1.10) | 0.40 |
| **Sex** | | | | | | |
| Male | 195 | 93 (47.69) | Ref | | | |
| female | 233 | 107 (45.92) | 0.96 (0.95-1.07) | 0.715 | 0.99 (0.93-1.05) | 0.79 |
| **Occupation** | | | | | | |
| Un employed | 255 | 108 (42.35) | Ref | | | |
| Self-employed | 109 | 59 (54.13) | 1.65 (1.02-2.52) | 0.360 | 2 (1.21-3.33) | 0.007 |
| Employed | 64 | 33 (50.09) | 1.45 (0.84-2.51) | 0.831 | 1.9 (1.04-3.41) | 0.036 |
| **Source of insulin & syringes** | | | | | | |
| At a health facility | 378 | 166 (43.92) | Ref | | | |
| Other sources | 50 | 34 (68.00) | 2.71 (1.45-5.09) | 0.001 | 2.51 (0.99-6.38) | 0.053 |
| **Injection site rotation** | | | | | | |
| Yes | 155 | 66 (42.58) | Ref | | | |
| No | 273 | 134 (49.08) | 1.30 (0.87-1.93) | 0.191 | 1.63 (1.03-2.32) | 0.037 |
| **Presence of comorbidity** | | | | | | |
| Yes | 102 | 40 (39.22) | 1.07 (0.99-1.14) | 0.073 | 1.03 (0.96-1.11) | 0.35 |
| No | 326 | 160 (49.08) | Ref | | | |
| **Insulin storage** | | | | | | |
| 2 to 8°C | 339 | 150 (44.25) | Ref | | | |
| >8°C | 88 | 49 (55.68) | 1.26 (0.86-1.01) | 0.061 | 0.95 (0.88-1.03) | 0.21 |
| **Blood glucose level** | | | | | | |
| Controlled | 88 | 31 (35.23) | Ref | | | |
| Uncontrolled | 340 | 169 (49.71) | 1.82 (1.12-2.96) | 0.015 | 1.69 (1.01-2.84) | 0.049 |

*The model was adjusted for insulin formulation, needle length, injection techniques, and demographic information (age, sex, occupation).*

the same effect size, with a minimal detectable effect of PR = 1.58 at 80% power. It is therefore possible that smaller, but clinically meaningful, risk factors for abscess formation were not identified, despite accounting for variance inflation from multiple covariates.

## Conclusions

This study indicates that skin complications at insulin injection sites are highly prevalent. The findings suggest that improper injection technique, poor site rotation, and uncontrolled blood glucose are significant modifiable associated factors. The use of insulin pens may help reduce the risk of abscesses.

## Recommendation

Targeted patient education on correct injection practices and glycemic control is crucial to minimize the burden of abscesse3s and scars at insulin injection site. However, given that the findings are observational in nature and due to the lack of statistical adjustments for some significant confounders, we advocate for longitudinal or controlled studies to substantiate causality.

## Acknowledgments

The authors sincerely thank the Muhimbili University of Health and Allied Sciences–Research and Ethics Committee (MUHAS-REC) for providing ethical clearance and the administrations of the visited hospitals (Mwananyamala, Temeke, Shiree Hindumandal and Kairuki) for providing permission for data collection. We also thank each appointed hospital HCP for their guidance during the data collection. Additionally, we thank the Department of Clinical Pharmacy and Pharmacology at Muhimbili University of Health and Allied Sciences for their support.

## Author contributions

**Conceptualization:** Dennis Patson Mbwambo, Wigilya Mikomangwa, Bertha Mally, Manase Kilonzi, Kaushik Ramaiya, Ritah Mutagonda, Alphonce Ignace Marealle.

**Data curation:** Dennis Patson Mbwambo.

**Formal analysis:** Dennis Patson Mbwambo.

**Funding acquisition:** Dennis Patson Mbwambo.

**Investigation:** Dennis Patson Mbwambo, Magreth Angelus.

**Methodology:** Dennis Patson Mbwambo, Mary Mayige, Alphonce Ignace Marealle.

**Project administration:** Dennis Patson Mbwambo.

**Resources:** Dennis Patson Mbwambo.

**Software:** Dennis Patson Mbwambo.

**Supervision:** Wigilya Mikomangwa, Bertha Mally, Kaushik Ramaiya, Mary Mayige, Ritah Mutagonda, Alphonce Ignace Marealle.

**Validation:** Dennis Patson Mbwambo.

**Writing – original draft:** Dennis Patson Mbwambo, Alphonce Ignace Marealle, Method Kazaura.

**Writing – review & editing:** Dennis Patson Mbwambo, Rajabu Hussein Mnkugwe, Manase Kilonzi, Magreth Angelus.

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
