## [Decision Letter · Decision Letter 0]

22 Apr 2025

Dear Dr. Mbwambo,

We look forward to receiving your revised manuscript.

Kind regards,

Dr. Mohammed Misbah Ul Haq, Pharm-D

Academic Editor

PLOS ONE

Journal Requirements:

For additional information about PLOS ONE ethical requirements for human subjects research, please refer to http://journals.plos.org/plosone/s/submission-guidelines#loc-human-subjects-research .

Reviewers' comments:

Reviewer's Responses to Questions

**Comments to the Author**

1. Is the manuscript technically sound, and do the data support the conclusions?

Reviewer #1: Yes

Reviewer #2: Yes

2. Has the statistical analysis been performed appropriately and rigorously?

Reviewer #1: I Don't Know

Reviewer #2: Yes

3. Have the authors made all data underlying the findings in their manuscript fully available?

Reviewer #1: Yes

Reviewer #2: Yes

4. Is the manuscript presented in an intelligible fashion and written in standard English?

Reviewer #1: Yes

Reviewer #2: No

Reviewer #1: The author should reorganize the manuscript’s structure and revise grammatical errors throughout the content. It may also be helpful for the editor to provide guidance on formatting. Additional comments are provided in the attached file.

Reviewer #2: Dear Authors

Congratulation on your hard and nice work.

My main concern is about linguistic issue of your manuscript ;

I recommend to you to re write the manuscript in standard English langue and correct all errors of it.

thank you very much

**Do you want your identity to be public for this peer review?** For information about this choice, including consent withdrawal, please see our Privacy Policy

Reviewer #1: No

Reviewer #2: **Yes:** Abdullah Saeed Abdullah

---

## [Author Response · Author response to Decision Letter 1]

26 May 2025

Dennis Patson Mbwambo

P.O.Box 6166

Dar Es Salaam

Tanzania

17th May 2025.

Dr. Mohammed Misbah Ul Haq, Pharm-D

Academic Editor

PLOS ONE

REF: Rebuttal to Review of the manuscript Titled “Skin-related adverse events and their determinants among Diabetic patients on insulin therapy in Tanzania”

Dear Dr. Misbah,

I am writing to respectfully address the queries recommended following the assessment of the above-mentioned manuscript. There were seven issues and each has been responded accordingly.

1. “The Inconsistent number: the author mentions 530 million people with DM in 2019 but later refer to 463 million. These figures should be consistent or clarified (perhaps 530 million is a more recent estimate?).

o Rebuttal: This has been addressed in paragraph 1 & 2 at the introduction part of the revised manuscript.

2. “When the author mentions, “in recent years,” or “just in two decades,” it is better to clarify the starting point if possible (e.g., “since 2000”).

o Rebuttal: The starting point has been added at paragraph 1 last sentence.

3. "Cross-section" → should be "cross-sectional". And "analytical cross-section study" → "analytical cross-sectional study".

o Rebuttal: Acknowledged and the word “cross-section” has been replaced with “cross-sectional” throughout the manuscript.

4. "Three-month random blood glucose readings" → "random" may be confusing in this context; better to clarify. For example, "random blood glucose values recorded over the past three months".

o Rebuttal: Acknowledged: the suggested amendments have been made as observed in the second-to-last sentence of the paragraph in the section on data collection procedures.

5. The phrase “we recruited one eligible subject after skipping another” is vague. Better phrasing: “Systematic sampling was used with a predetermined interval (every second eligible patient) until the required sample size per facility was reached.

o Rebuttal: Acknowledged, the changes have been made as observed in paragraph 2 of the sample size and sampling technique section.

6. Redundant and Disorganized Presentation: Some key findings (e.g., syringe reuse and insulin source) are repeated across multiple paragraphs. The flow between patient characteristics, prevalence data, and factors associated with outcomes lacks structure, which may confuse readers and dilute key messages

o Rebuttal: Acknowledged. The repetitive sentences have been removed, and the work has been rearranged to provide a suitable flow. This can be observed from the demographic information to factors associated with the outcomes of interest.

7. Language and Formatting Issues: There are several grammatical errors, awkward phrases (“per a needle”), and inconsistent formatting of numbers and percentages (e.g., missing spaces or switching between formats). These stylistic issues reduce clarity and compromise the scientific tone of the section.

o Rebuttal: Acknowledged, Language and Formatting Issues have been addressed throughout the manuscript.

Supporting Documentation

I have attached the following for your review:

• A marked-up copy of the revised manuscript.

• A revised manuscript without mark-up

Request for Reconsideration

I kindly ask that you reevaluate the revised manuscript in light of this clarified information. I am open to further discussion or revisions to resolve any remaining concerns.

Thank you for your time and attention to this matter. Please contact me at +255658333469/dennis.mbwambo@gmail.com with any questions.

Sincerely,

Denis Patson Mbwambo

The auther.

---

## [Decision Letter · Decision Letter 1]

10 Jun 2025

Dear Dr. Mbwambo,

We look forward to receiving your revised manuscript.

Kind regards,

Dr. Mohammed Misbah Ul Haq, Pharm-D

Academic Editor

PLOS ONE

Reviewers' comments:

Reviewer's Responses to Questions

**Comments to the Author**

Reviewer #3: (No Response)

Reviewer #4: (No Response)

2. Is the manuscript technically sound, and do the data support the conclusions?

Reviewer #3: Partly

Reviewer #4: Partly

3. Has the statistical analysis been performed appropriately and rigorously?

Reviewer #3: No

Reviewer #4: No

4. Have the authors made all data underlying the findings in their manuscript fully available?

Reviewer #3: No

Reviewer #4: No

5. Is the manuscript presented in an intelligible fashion and written in standard English?

Reviewer #3: No

Reviewer #4: No

Reviewer #3: General comments

This is an interesting study addressing an important public health issue, highlighting a substantial burden of skin-related complications among diabetic patients receiving insulin therapy. The findings are valuable and have clear implications for clinical practice and health policy in Tanzania. Overall, the manuscript presents original research with findings relevant to its specific context.

However, there are several points that would benefit from clarification or revision to enhance rigor, clarity, and transparency.

Referencing and language

1. Several references appear mismatched or irrelevant. For example, reference #3 ("Tennessee Department of Education") seems unrelated to the manuscript's topic. Similarly, reference #1 ("An audit of insulin usage and insulin injection practices in a large Indian cohort") does not clearly support the global statement regarding the increasing prevalence of diabetes mellitus (DM). Additionally, references 3 and 4 are cited after references 5 and 6, disrupting the logical flow of the citations. Some factual claims, such as "it was estimated that 530 million individuals globally were living with DM," currently lack references. After a brief review of the reference list, similar issues were observed elsewhere; it would thus be valuable to carefully verify the citations and ensure each reference is relevant and correctly placed. For instance, clarify the relevance of referencing an online article from MedicalNewsToday ("Which skin conditions are linked to type 2 diabetes?").

2. The manuscript would also greatly benefit from careful language editing to ensure consistent and high-quality academic English throughout the text.

Abstract

3. The abstract currently provides detailed statistical information (specific prevalence ratios, confidence intervals, and p-values). Consider if this level of statistical detail is necessary in the abstract, as it may reduce readability and accessibility. A more concise summary of key significant findings might be more suitable.

4. Additionally, the abstract's conclusion currently presents strong policy recommendations regarding training interventions. Given the observational, cross-sectional study design, and the absence of adjustments for potential confounding factors, please consider moderating this to clearly reflect that your findings are preliminary and observational in nature. It may be more appropriate, at least as the manuscript currently stands, to emphasize the need for further research rather than making immediate policy recommendations, particularly given that no intervention regarding training or education was directly evaluated within this study.

Background

5. You state: "In recent years, since 2017, it was observed that insulin consumption has increased proportionally with the increased prevalence of DM (3,8)." The need for multiple references to support this fairly general observation seems unclear. Consider simplifying or explicitly justifying the multiple references.

6. Additionally, rather than focusing primarily on the lack of studies in low-income countries, it could be more beneficial to clearly link the observed global trend of increased insulin use directly to your study’s objective. For instance, highlighting how increased insulin use might particularly affect low- and middle-income countries such as Tanzania could strengthen the rationale behind your research.

Materials and Methods

7. The reference provided to support the statement that Dar es Salaam is Tanzania's largest city ("https://sensa.nbs.go.tz/publication/volume1a.pdf") appears unnecessary, as this is widely acknowledged general knowledge.

8. Clarifying the inclusion criteria would improve transparency, particularly explaining why the minimum duration of insulin therapy was set at four weeks, as this choice currently seems arbitrary.

9. An important methodological issue concerns your exclusion of patients with pre-existing abscesses or scars at the initiation of insulin therapy. Due to this criterion, the study effectively assesses the incidence of new skin-related adverse events rather than overall prevalence. This distinction should be explicitly clarified in your methods and briefly discussed in both results and discussion sections, as your presented prevalence figures likely underestimate the true burden. Moreover, since abscesses and scars were assessed retrospectively (over the past 12 months), it remains unclear how this exclusion criterion was practically implemented without introducing selection bias. Additionally, this exclusion criterion may not be strictly necessary, since abscesses and scars arising after the initiation of insulin therapy could potentially have causes other than insulin injections. Clearly describe the practical implementation of your inclusion and exclusion criteria, justify the rationale behind them, and explicitly address potential biases resulting from this methodological choice.

10. You mention using systematic sampling by selecting every second eligible patient. Clarifying or briefly justifying this particular sampling interval would enhance methodological transparency, especially since this interval does not appear to reflect a commonly used standard.

11. In the description of the case report form (CRF), considerable attention is given to its development and pretesting. It would be more valuable to clearly and explicitly define key variables and measurements used in the study—such as criteria for abscess and scar classification, needle reuse practices, insulin storage conditions, and glycemic control—rather than extensively describing the CRF development itself.

12. You state that "random blood glucose values" were used to define glycemic control status, but provide no details regarding how these values were operationalized or interpreted to categorize control status (e.g., thresholds used, number or frequency of measurements). Additionally, explicitly justify why HbA1c, a standard measure of long-term glycemic control, was not used. Clarifying this is crucial for reproducibility.

13. The manuscript also mentions data collected from patient files and "during follow-up." Since your study is cross-sectional, the term "follow-up" is potentially misleading, suggesting a longitudinal design. Please explicitly clarify what is meant here.

14. You have used modified Poisson regression, which is appropriate for estimating prevalence ratios. However, the apparent lack of adjustment for potential confounders is methodologically problematic. Without controlling for relevant covariates, such as age, sex, duration of diabetes, insulin dose, BMI, or socioeconomic factors, the observed associations between injection techniques or glucose control and skin complications might be confounded. Consider performing sensitivity analyses or additional modeling that incorporates relevant covariates, or at minimum clearly acknowledge this as a major limitation in your discussion.

Results and Discussion

15. In the results section, consider briefly summarizing the key significant findings in the text and avoiding overly detailed numerical repetition, as these details already appear fully in the tables.

16. Regarding Table 3, the rationale for reporting abscess and scar occurrences in quarterly intervals is currently unclear. Consider clarifying why these intervals were selected and their relevance for interpreting the findings.

17. The results have inconsistencies between tables and the descriptive text. In the main text, you state that those who were self-employed had a 134% higher prevalence of abscess formation compared to the non-employed, with an adjusted prevalence ratio (aPR) of 2.34 (95% CI: 1.04–5.27) and a reported p-value of 0.4. However, Table 4 lists the corresponding p-value as 0.04, consistent with the stated confidence interval. This discrepancy suggests a typographical error in the main text.

18. Additionally, in Table 5 ("Source of insulin & syringes"), the adjusted prevalence ratio (aPR) for scar formation is given as 2.51, but the confidence interval (1.22–1.98) does not logically correspond to this estimate, as it does not encompass the stated point estimate. Similarly, Table 4 includes further problematic confidence intervals: for the insulin pen formulation, the crude prevalence ratio is given as 0.19 (CI: 0.23–0.93), which is incorrect as the lower bound exceeds the point estimate. Likewise, the crude prevalence ratio for needle length "≤ 6mm" is stated as 3.30 (CI: 1.01–1.05), an interval clearly inconsistent with the stated point estimate. Additionally, the category "Unknown needle length" shows a crude prevalence ratio of 2.60 with a confidence interval (0.85–0.98) that is entirely below the given point estimate. Taken together, these inconsistencies strongly suggest typographical or transcription errors in reporting your statistical results. It is recommended that you thoroughly verify and correct all numerical values throughout the text and tables to ensure accurate and consistent reporting of your findings.

19. Given your exclusion criterion (patients with pre-existing abscesses or scars at insulin initiation), explicitly reflect on how this might introduce selection bias into your prevalence estimates. This methodological limitation significantly affects the interpretation of your results, as it may shift your reported figures towards incidence estimates rather than true prevalence. Please clarify and discuss the implications of this choice for the validity and generalizability of your findings.

20. Additionally, you describe a "170% higher prevalence" associated with improper injection techniques. While this figure is technically correct, it might be clearer and easier for readers to interpret if you restate this as a "2.7-fold higher prevalence" or "approximately 2.7 times higher prevalence." Ensure consistent and clear use of prevalence ratios and wording throughout the manuscript to avoid confusion.

21. The wording in the discussion (e.g. "can lead to abscess") implies causality regarding the association between improper insulin injection techniques and abscess formation. Given the cross-sectional design and the lack of statistical adjustments for confounding variables, causal inference should be avoided. Please explicitly adjust your language to reflect unadjusted associations rather than causal relationships.

22. You also note an association between poorly controlled blood glucose and scar formation but provide insufficient explanatory context ("not a new phenomenon").

23. The conclusions are clearly formulated and highlight important practical recommendations based on your findings. However, the current wording implies causal interpretations of your results (e.g., describing abscesses as directly associated with improper injection site rotation and insulin pens as "protective"). Given the cross-sectional design, the absence of statistical adjustment for potential confounders, and the methodological limitations already discussed, your conclusions should explicitly avoid suggesting causality. Instead, clearly state that the identified associations are observational and preliminary, emphasizing the need for longitudinal or controlled studies to confirm causality before making firm policy recommendations. Alternatively, if you argue that your results replicate previously established causal relationships from prior research, you must explicitly demonstrate and discuss this replication more thoroughly within your discussion section.

24. Finally, the manuscript would greatly benefit from a dedicated section explicitly describing the study’s limitations, clearly stating the cross-sectional design, potential recall bias, measurement issues (particularly regarding glycemic control), and concerns about generalizability.

Thank you for the opportunity to read and review your interesting work, I hope these comments will be helpful as you finalize your manuscript. Best of luck with your research!

Reviewer #4: The study appears to address a relevant gap in knowledge regarding skin-related adverse events in diabetic patients on insulin therapy in a low-to-middle income country setting. The cross-sectional design is appropriate for assessing prevalence and associated factors. However, the manuscript should explicitly address potential limitations of this design in the discussion, particularly regarding causality. While the study aims to identify determinants, the cross-sectional nature limits the ability to establish temporal relationships.

The sample size and selection of hospitals in Dar es Salaam should be justified more thoroughly. Were the hospitals selected representative of the broader healthcare landscape in the region? A power analysis justifying the sample size would strengthen the methodology.

The manuscript needs to specify the statistical methods used to analyze the data in more detail. What specific regression models were employed to identify determinants? How were potential confounders addressed in the analysis?

The manuscript should describe how missing data were handled in the analysis.

Modified Poisson regression is suitable for binary outcomes with high prevalence. Sample size calculation (n=428) is justified (95% CI, 5% precision, 50% prevalence assumption). Determinants were identified via modified Poisson regression, but Table 4/5 show incomplete adjustment (e.g., occupation was adjusted, but age/sex were not consistently included in final models).

Confounding: Uncontrolled glucose is both a determinant and potential consequence of poor injection practices. Temporal ambiguity weakens causal inferences.

Sample representativeness: Limited to 4 hospitals in Dar es Salaam; rural settings not included.

Occupational association: Self-employment linked to abscesses/scars (aPR=2.34, p=0.04; aPR=2.0, p=0.007), but no mechanistic exploration (e.g., type of work, hygiene access).

Verdict: Partially sound. Data broadly support conclusions, but causal claims are limited by design and analysis gaps.

Variables with p<0.2 in univariate analysis were included in multivariate models, but key covariates (e.g., age, sex) were often non-significant and dropped. Final models lack adjustment for basic demographics.

Table 4 lists "adjusted aPR" but includes unadjusted results (e.g., "Source of insulin" lacks adjusted estimates). Confidence intervals for some aPRs are implausible (e.g., aPR=2.51; 95% CI: 1.22–1.98 for scar determinants).

Missing details: No rationale for using Poisson over logistic regression; no sensitivity analyses.

Verdict: Moderately rigorous. Analysis requires refinement for robust inference.

There are instances of awkward phrasing and grammatical errors throughout the manuscript. A thorough proofread and editing by someone with strong English language skills is recommended. For example, [Suggested Revision in Paragraph 2 in the introduction: "This problem is more pronounced in developing countries where resources are limited, and the number of people with diabetes is increasing at an alarming rate." or, even better] or [A hospital-based cross-sectional study design was employed..."], and other typos such as "upper hand" (p.25); "musk" (p.33); "shorts" (Table 3) etc. Grammar (e.g., "We noted" vs. "We observe").

**Do you want your identity to be public for this peer review?** For information about this choice, including consent withdrawal, please see our Privacy Policy

Reviewer #3: No

Reviewer #4: No

---

## [Author Response · Author response to Decision Letter 2]

25 Jul 2025

Reviewer #3:

Referencing and language

1. Several references appear mismatched or irrelevant. For example, reference #3 ("Tennessee Department of Education") seems unrelated to the manuscript's topic. Similarly, reference #1 ("An audit of insulin usage and insulin injection practices in a large Indian cohort") does not clearly support the global statement regarding the increasing prevalence of diabetes mellitus (DM). Additionally, references 3 and 4 are cited after references 5 and 6, disrupting the logical flow of the citations. Some factual claims, such as "it was estimated that 530 million individuals globally were living with DM," currently lack references. After a brief review of the reference list, similar issues were observed elsewhere; it would thus be valuable to carefully verify the citations and ensure each reference is relevant and correctly placed. For instance, clarify the relevance of referencing an online article from MedicalNewsToday ("Which skin conditions are linked to type 2 diabetes?").

Rebuttal: Referencing issue has been adressed.

Documentation: Introduction and pages 16-18 (discussion)

2. The manuscript would also greatly benefit from careful language editing to ensure consistent and high-quality academic English throughout the text.

Rebuttal: Language editing has been done

Documentation: All pages of the manuscript

Abstract

3. The abstract currently provides detailed statistical information (specific prevalence ratios, confidence intervals, and p-values). Consider if this level of statistical detail is necessary in the abstract, as it may reduce readability and accessibility. A more concise summary of key significant findings might be more suitable.

Rebuttal: Findings has been summarized on the abstract section.

Documentation: Page 2 (result section)

4. Additionally, the abstract's conclusion currently presents strong policy recommendations regarding training interventions. Given the observational, cross-sectional study design, and the absence of adjustments for potential confounding factors, please consider moderating this to clearly reflect that your findings are preliminary and observational in nature. It may be more appropriate, at least as the manuscript currently stands, to emphasize the need for further research rather than making immediate policy recommendations, particularly given that no intervention regarding training or education was directly evaluated within this study.

Rebuttal: Abstracti conclusion has been moderated.

Documentation: Page 3 (conclusion)

Background

5. You state: "In recent years, since 2017, it was observed that insulin consumption has increased proportionally with the increased prevalence of DM (3,8)." The need for multiple references to support this fairly general observation seems unclear. Consider simplifying or explicitly justifying the multiple references.

Rebuttal: The unnecessary multile referencing has been adressed.

Documentation: Page 2 (conclusion)

6. Additionally, rather than focusing primarily on the lack of studies in low-income countries, it could be more beneficial to clearly link the observed global trend of increased insulin use directly to your study’s objective. For instance, highlighting how increased insulin use might particularly affect low- and middle-income countries such as Tanzania could strengthen the rationale behind your research.

Rebuttal: Query has been addressed.

Documentation: Page 3 (Background)

Materials and Methods

7. The reference provided to support the statement that Dar es Salaam is Tanzania's largest city ("https://sensa.nbs.go.tz/publication/volume1a.pdf") appears unnecessary, as this is widely acknowledged general knowledge.

Rebuttal: The unnecessary reference has been removed.

Documentation: Page 4 (Study design and setting)

8. Clarifying the inclusion criteria would improve transparency, particularly explaining why the minimum duration of insulin therapy was set at four weeks, as this choice currently seems arbitrary.

Rebuttal: The inclusion criteria has been clarified.

Documentation: Page 5 (Study population)

9. An important methodological issue concerns your exclusion of patients with pre-existing abscesses or scars at the initiation of insulin therapy. Due to this criterion, the study effectively assesses the incidence of new skin-related adverse events rather than overall prevalence. This distinction should be explicitly clarified in your methods and briefly discussed in both results and discussion sections, as your presented prevalence figures likely underestimate the true burden.

Rebuttal: The methodological issue(exclussion of those with pre-existing abscesses and scars) has been clarified in methods and discussed in both results and discussion sections.

Documentation: Page 5, and page 16.

Moreover, since abscesses and scars were assessed retrospectively (over the past 12 months), it remains unclear how this exclusion criterion was practically implemented without introducing selection bias. Additionally, this exclusion criterion may not be strictly necessary, since abscesses and scars arising after the initiation of insulin therapy could potentially have causes other than insulin injections. Clearly describe the practical implementation of your inclusion and exclusion criteria, justify the rationale behind them, and explicitly address potential biases resulting from this methodological choice.

Rebuttal: The same has been adressed (exclussion of those with pre-existing abscesses and scars)

Documentation: Page 5, and page 16.

10. You mention using systematic sampling by selecting every second eligible patient. Clarifying or briefly justifying this particular sampling interval would enhance methodological transparency, especially since this interval does not appear to reflect a commonly used standard.

Rebuttal: Sampling interval has been clarified.

Documentation: Page 5(Sampling technique).

11. In the description of the case report form (CRF), considerable attention is given to its development and pretesting. It would be more valuable to clearly and explicitly define key variables and measurements used in the study—such as criteria for abscess and scar classification, needle reuse practices, insulin storage conditions, and glycemic control—rather than extensively describing the CRF development itself.

Rebuttal: Key variables and measurements used in the study have been defined.

Documentation: Page 6 & 7.

12. You state that "random blood glucose values" were used to define glycemic control status, but provide no details regarding how these values were operationalized or interpreted to categorize control status (e.g., thresholds used, number or frequency of measurements). Additionally, explicitly justify why HbA1c, a standard measure of long-term glycemic control, was not used. Clarifying this is crucial for reproducibility.

Rebuttal: Query regarding the glycemic control status measurement has been adressed.

Documentation: Table No1, Page 7.

13. The manuscript also mentions data collected from patient files and "during follow-up." Since your study is cross-sectional, the term "follow-up" is potentially misleading, suggesting a longitudinal design. Please explicitly clarify what is meant here.

Rebuttal: Data collection procedure using information from patient file has been clarified and the confusing word “follow-up” has been adressed.

Documentation: Table No1, Page 8.

14. You have used modified Poisson regression, which is appropriate for estimating prevalence ratios. However, the apparent lack of adjustment for potential confounders is methodologically problematic. Without controlling for relevant covariates, such as age, sex, duration of diabetes, insulin dose, BMI, or socioeconomic factors, the observed associations between injection techniques or glucose control and skin complications might be confounded. Consider performing sensitivity analyses or additional modeling that incorporates relevant covariates, or at minimum clearly acknowledge this as a major limitation in your discussion.

Rebuttal: Detailed information regarding the use of the modified Poisson regression and the adjustment of potential con-founders has been provided. Either the limitations for uncaptured possible counfouders has been acknowledged in relevant sections.

Documentation: Page 9(Data analysis), page 18(limitation) and page 19 (conclusion).

Results and Discussion

15. In the results section, consider briefly summarizing the key significant findings in the text and avoiding overly detailed numerical repetition, as these details already appear fully in the tables.

Rebuttal: Results has been briefly summarized.

Documentation: Page 1-18 (Results and discussion).

16. Regarding Table 3, the rationale for reporting abscess and scar occurrences in quarterly intervals is currently unclear. Consider clarifying why these intervals were selected and their relevance for interpreting the findings.

Rebuttal: Prevalence of abscesses and scars reporting style has been rectified.

Documentation: Page 12.

17. The results have inconsistencies between tables and the descriptive text. In the main text, you state that those who were self-employed had a 134% higher prevalence of abscess formation compared to the non-employed, with an adjusted prevalence ratio (aPR) of 2.34 (95% CI: 1.04–5.27) and a reported p-value of 0.4. However, Table 4 lists the corresponding p-value as 0.04, consistent with the stated confidence interval. This discrepancy suggests a typographical error in the main text.

Rebuttal: Results presented in tables and the descriptive text are now inconsistent.

Documentation: Tables for results and the descriptive information.

18. Additionally, in Table 5 ("Source of insulin & syringes"), the adjusted prevalence ratio (aPR) for scar formation is given as 2.51, but the confidence interval (1.22–1.98) does not logically correspond to this estimate, as it does not encompass the stated point estimate. Similarly, Table 4 includes further problematic confidence intervals: for the insulin pen formulation, the crude prevalence ratio is given as 0.19 (CI: 0.23–0.93), which is incorrect as the lower bound exceeds the point estimate. Likewise, the crude prevalence ratio for needle length "≤ 6mm" is stated as 3.30 (CI: 1.01–1.05), an interval clearly inconsistent with the stated point estimate. Additionally, the category "Unknown needle length" shows a crude prevalence ratio of 2.60 with a confidence interval (0.85–0.98) that is entirely below the given point estimate. Taken together, these inconsistencies strongly suggest typographical or transcription errors in reporting your statistical results. It is recommended that you thoroughly verify and correct all numerical values throughout the text and tables to ensure accurate and consistent reporting of your findings.

Rebuttal: Typos have been rectified in all relevant sections and tables.

Documentation: Tables 5 & 6. and the corresponding discussion section.

19. Given your exclusion criterion (patients with pre-existing abscesses or scars at insulin initiation), explicitly reflect on how this might introduce selection bias into your prevalence estimates. This methodological limitation significantly affects the interpretation of your results, as it may shift your reported figures towards incidence estimates rather than true prevalence. Please clarify and discuss the implications of this choice for the validity and generalizability of your findings.

Rebuttal: Query has been addressed in above responses.

Documentation: Discussion and limitation section.

20. Additionally, you describe a "170% higher prevalence" associated with improper injection techniques. While this figure is technically correc zt, it might be clearer and easier for readers to interpret if you restate this as a "2.7-fold higher prevalence" or "approximately 2.7 times higher prevalence." Ensure consistent and clear use of prevalence ratios and wording throughout the manuscript to avoid confusion.

Rebuttal: The reporting style has been addressed for clearer interpretation..

Documentation: Tables 5 & 6. and the corresponding discussion section.

21. The wording in the discussion (e.g. "can lead to abscess") implies causality regarding the association between improper insulin injection techniques and abscess formation. Given the cross-sectional design and the lack of statistical adjustments for confounding variables, causal inference should be avoided. Please explicitly adjust your language to reflect unadjusted associations rather than causal relationships.

Rebuttal: The incorrect word “can lead to abscess” which is misleading, has been rectified.

Documentation: Discussion section.

22. You also note an association between poorly controlled blood glucose and scar formation but provide insufficient explanatory context ("not a new phenomenon").

Rebuttal: Detailed and clear information has been provide.

Documentation: Discussion section.

23. The conclusions are clearly formulated and highlight important practical recommendations based on your findings. However, the current wording implies causal interpretations of your results (e.g., describing abscesses as directly associated with improper injection site rotation and insulin pens as "protective"). Given the cross-sectional design, the absence of statistical adjustment for potential confounders, and the methodological limitations already discussed, your conclusions should explicitly avoid suggesting causality. Instead, clearly state that the identified associations are observational and preliminary, emphasizing the need for longitudinal or controlled studies to confirm causality before making firm policy recommendations. Alternatively, if you argue that your results replicate previously established causal relationships from prior research, you must explicitly demonstrate and discuss this replication more thoroughly within your discussion section.

Rebuttal: The word “directly associated with” which implies causal interpretations has been removed I all relevant sections including at the conclusion part.

Documentation: The conclusion part.

24. Finally, the manuscript would greatly benefit from a dedicated section explicitly describing the study’s limitations, clearly stating the cross-sectional design, potential recall bias, measurement issues (particularly regarding glycemic control), and concerns about generalizability.

Rebuttal: The section for limitation has been added with all limitation admitted.

Documentation: Page 18. Limitations.

Reviewer #4: The study appears to address a relevant gap in knowledge regarding skin-related adverse events in diabetic patients on insulin therapy in a low-to-middle income country setting. The cross-sectional design is appropriate for assessing prevalence and associated factors. However, the manuscript should explicitly address potential limitations of this design in the discussion, particularly regarding causality. While the study aims to identify determinants, the cross-sectional nature limits the ability to establish temporal relationships.

Rebuttal: The section for limitation has been added with all limitation admitted.

Documentation: Page 18. Limitations.

The sample size and selection of hospitals in Dar es Salaam should be justified more thoroughly. Were the hospitals selected representative of the broader healthcare landscape in the region? A power analysis justifying the sample size would strengthen the methodology.

Rebuttal: The sample size and selection of hospitals in Dar es Salaam should has been justified. And A power analysis justifying the sample size has been captured.

Documentation: Page 4 and 9.

The manuscript needs to specify the statistical methods used to analyze the data in more detail. What specific regression models were employed to identify determinants? How were potential confounders addressed in the analysis?

The manuscript should describe how missing data were handled in the analysis.

Rebuttal: The the statistical methods used to analyze the data has been clarified in detail including the specific models employed to identify the associated factor.

Documentation: Page 9.

Modified Poisson regression is suitable for binary outcomes with

---

## [Decision Letter · Decision Letter 2]

10 Sep 2025

Dear Dr. Mbwambo,

We look forward to receiving your revised manuscript.

Kind regards,

Mohammed Misbah Ul Haq, Pharm-D

Academic Editor

PLOS ONE

Journal Requirements:

Reviewer's Responses to Questions

**Comments to the Author**

Reviewer #3: (No Response)

Reviewer #5: (No Response)

2. Is the manuscript technically sound, and do the data support the conclusions?

Reviewer #3: Yes

Reviewer #5: No

3. Has the statistical analysis been performed appropriately and rigorously?

Reviewer #3: Yes

Reviewer #5: No

4. Have the authors made all data underlying the findings in their manuscript fully available?

Reviewer #3: Yes

Reviewer #5: (No Response)

5. Is the manuscript presented in an intelligible fashion and written in standard English?

Reviewer #3: No

Reviewer #5: No

Reviewer #3: Thank you for addressing many of the points from the previous round. I can see substantial effort in refining the abstract, clarifying the four week inclusion rationale, adding a limitations paragraph, and correcting several reference mismatches. The manuscript has certainly improved. Below I outline the main items that, in my view, still merit attention before the paper will be methodologically solid and fully comprehensible to readers.

Methods

1. Sampling interval: you now describe k = 2 systematic sampling (every other patient). A single sentence confirming that the first patient was chosen at random in each clinic session would satisfy the assumption of systematic sampling.

2. Power analysis: thank you for adding post hoc calculations. It would be clearer to move these numbers to the Limitations paragraph and simply note in Methods that post hoc power was estimated.

3. Table 1 now states three random BG readings define “controlled.” Please add a short sentence explaining why this surrogate was preferred over fasting BG and how often readings were actually available (completeness).

Results

4. Provide the list of covariates included in each multivariable model at the foot of Tables 5 and 6 (e.g., “Model adjusted for age, sex, occupation, insulin formulation, comorbidity and blood glucose control”).

5. Several confidence interval bounds do not include the point estimate or are reversed. This suggests residual transcription errors. E.g. Table 5: age group ≥ 45 y crude PR 0.51 with CI 1.03–1.14; sex crude PR 0.93 with CI 0.97–1.06; Table 6: age 25 44 crude PR 1.07 with CI 0.89–1.06.

6. Where crude PRs reverse direction after adjustment (e.g., insulin formulation), flag this in a sentence so readers understand the role of confounding.

Discussion

7. The new comparisons with Ethiopia, India and Bangladesh are helpful. Where your prevalence is higher or lower, add a short clause on possible drivers (clinical training standards, availability of pens, sample age).

8. The paragraph on insulin pen “protective advantage” still reads causal. Please rephrase so that the finding is framed as an observed association in your dataset. You may note that similar associations from Saudi Arabia and Lebanon increase the plausibility of a causal link—but, unless those papers explicitly demonstrate causality (I have not verified this), they should be cited only as supportive observational evidence. At the same time, acknowledge that residual confounding (for example, socioeconomic status or prior injection technique training) might partly explain the pattern. If robust longitudinal or interventional studies do show a causal benefit of pen devices, consider citing them; otherwise, emphasize that prospective confirmation is still needed.

9. Discussion still contain causal verbs (“resulted in”, “led to”, “confer a protective advantage”). Because the study is cross sectional, please replace with associative phrasing (“was associated with”, “showed lower prevalence”) or explicitly frame any causal interpretation as speculative.

10. Strengthen the limitations bullet on measurement error. Random BG values can be influenced by recent meals; briefly state that misclassification could bias associations toward or away from null.

Miscellaneous

11. Remove the repeated WHO pharmacovigilance reference (#10 and #16 appear identical) .

12. Typographic clean up: “Forexample” → “For example”; double periods after citations (#8., #9…) to single period; etc.

Most earlier comments have been addressed, but the numerical inconsistencies in several confidence intervals, residual causal language, and incomplete reporting of adjustment variables still need resolution. Clarifying these points will ensure that readers (and potential policy users) can interpret the findings with confidence.

I appreciate the authors’ responsiveness and look forward to a cleaner, internally consistent final version.

Reviewer #5: GENERAL COMMENTS/QUERIES/RECOMMENDATIONS:

1. The manuscript describes important clinical phenomena to all practicing endocrinologists/physicians the world over. If done correctly, it would contribute to otherwise unknown quantitative estimates of observed clinical phenomena in African settings.

2. There are several flawed grammatical, syntactic, technical as well as logical attributes throughout the document as shown in the specific comments the underneath.

e.g. since the entire study had its gravity on ‘skin adverse events’, that are by default unfavourable and rare events, and almost always sporadic (rather than prevalent!), didn’t investigators/authors consider the need for the study design to have been ‘prospective observational clinical research’ by design?

Otherwise, why doesn’t the manuscript have page numbers?

Specific comments/queries/recommendations:

1. Conclusion section of a manuscript ought to summarise what was found in the study without repeating any figures/estimates. In this manuscript, some sentences in the conclusion section seem to belong to recommendations section rather than conclusion section.

Evidence: “Consequently, we advocate for longitudinal or controlled studies to ascertain causality”. (refer conclusion sub-section of the abstract sction in the manuscript)

2. Some of the findings in the abstract’s results’ sub-section have their quantitative estimates omitted, and hence losing the meaning of the findings presentation.

e.g. why were the positive associated factors for scars formation not accompanied by their statistical estimates vivid in the multivariable section of the results section in the main text?

Note: in most cases, it is the abstract section that will ever be read in future, once the article becomes published. As a summary of the entire manuscript, it still needs to be as informative as possible!

3. It is important for the authors to be meticulous with the usage of technical words! At present, I had difficulties to believe if authors knew how to differentiate ‘adverse events’ from ‘side effects’ when it comes to pharmacological management of diabetes mellitus.

Reason: whereas diabetes keto-acidosis may be considered as an ‘adverse event’ among people with diabetes mellitus consuming oral hypoglycaemics (refer to the example provided by FDA adverse event reporting), the same ‘diabetes keto-acidosis’ is customarily regarded as a ‘side effect’ of insulin treatment among people with diabetes mellitus.

Recommendations: Authors need to consider another ‘adverse event(s)’ reported in literature that specifically addresses insulin usage among diabetics when building their argument about ‘why the study was done’

4. Can we confidently justify this study to had been a CROSS-SECTIONAL by design during the stipulated timeline?

Reason: Based on what authors have provided in the manuscript’s methods section, there are clear indications that the study was RETROSPECTIVE by design on a de-facto basis!!!

Evidence:

“Data were meticulously gathered from both patient files and direct interviews. The data extracted from patient records encompassed random blood glucose readings, sociodemographic details, and other pertinent information.”

(refer to 2nd paragraph in the sub-section data collection procedure)

And again

“In addition to the physical examination for the identification of existing abscesses and scars, patients were queried about the development of any abscesses or scars within the twelve months preceding the interview.”

Recommendations: I would kindly request authors to either consult a clinical epidemiologist or simply refer to any basic epidemiology textbook that may provide a description for the differences between the two study designs.

5. The description of how variables were treated before and during data analysis is missing in the manuscript

e.g. why was the variable ‘age of participants’ categorized? And maybe more interesting, what was the clinical/logistical basis of the cut-off age groups? (refer to table 2 in the results section for the logic behind the query!)

Reason: Do authors understand that categorizing an otherwise naturally occurring continuous variable during data analysis is associated with some loss of information? That in turn may render the variable ‘not significant’ (type II error), especially if the study was under-powered??

6. I am still doubtful if the usage of ‘modified Poisson regression model’ was appropriate in assessing factors associated with both abscess and scar formation in this study findings.

Evidence: By default, poisson regression (and its modified versions) are reserved for otherwise ‘rare events’. The fact that in this study they have reported the prevalence of scars formation to have been 46.7% and that of abscess as 22.2% rule-out the possibility for the data to fit in poisson regression model!

Reason: can one justify a prevalence rate of 46.7% to be a rare event?

Recommendations: Authors need to consult a ‘chartered statistician’ or a ‘qualified biostatistician’ for assistance into the matter. Likewise, in their subsequent re-submission, authors ought to make effort to attach their ‘minimalised dataset’ as well for reviewers’ verification(s).

7. Why do some estimates reported to be 'outside the bound' of 95% confidence interval?

Evidence: refer to age group ≥ 45 years aPR and its corresponding 95% C.I. in table 5 (aPR=0.6, 95% C.I.: 0.99 – 1.15) !!!

Note: it is highly probable that there was profound model misspecification and therefore entire thinking of fitting a POISSON REGRESSION model was invalid in this study!

Recommendations: Consult a ‘chartered statistician’ or a qualified biostatistician for urgent assistance on the matter.

8. Authors are advised to stick to ‘standards of scientific writing’ in their manuscript.

Reason: interpretation of findings is customarily reserved to discussion section of the manuscript. It was surprising to see interpretation of findings made in the results section of this manuscript!

Evidence: Furthermore, those employing insulin pens exhibited a prevalence 0.13 times lower when compared to

individuals utilising vials and syringes (aPR = 0.13; 95%CI: 0.04-0.48; p = 0.002)

9. There is evidence of vivid erroneous interpretation of analysed findings in this manuscript!

e.g. Furthermore, those employing insulin pens exhibited a prevalence 0.13 times lower when compared to individuals utilising vials and syringes (aPR = 0.13; 95%CI: 0.04-0.48; p = 0.002)

Why it was wrong? – when reporting how much lower for cases where the estimates are below a unit, the proper interpretation is to consider the difference from the unit

Recommendations: should authors consider to report the way they have reported (highly recommended for that to appear in the DISCUSSION section and not where it has been cited in text!) then, they ought to write

“ ..exhibited a a prevalence of 87 times lower when compared to…”

Reason: the true relative lower value being = 100-13.

**Do you want your identity to be public for this peer review?** For information about this choice, including consent withdrawal, please see our Privacy Policy

Reviewer #3: No

Reviewer #5: **Yes:** Kelvin Melkizedeck Leshabari

---

## [Author Response · Author response to Decision Letter 3]

21 Oct 2025

Responded by an attachment document

---

## [Decision Letter · Decision Letter 3]

19 Jan 2026

Skin-related adverse events and their associated factors among Diabetic patients on insulin therapy.

PONE-D-25-08926R3

Dear Dr. Mbwambo,

We’re pleased to inform you that your manuscript has been judged scientifically suitable for publication and will be formally accepted for publication once it meets all outstanding technical requirements.

Kind regards,

Dr. Mohammed Misbah Ul Haq, Pharm-D

Academic Editor

PLOS One

Additional Editor Comments (optional):

Reviewers' comments:

Reviewer's Responses to Questions

**Comments to the Author**

Reviewer #5: (No Response)

2. Is the manuscript technically sound, and do the data support the conclusions?

Reviewer #5: Yes

3. Has the statistical analysis been performed appropriately and rigorously?

Reviewer #5: Yes

4. Have the authors made all data underlying the findings in their manuscript fully available?

Reviewer #5: Yes

5. Is the manuscript presented in an intelligible fashion and written in standard English?

Reviewer #5: (No Response)

Reviewer #5: Response:

1. As queried before, some of the findings and findings interpretations in the abstract section are still erroneous!

e.g. for insulin pen usage and its association with abscess formation, the estimated aPR was reported as 0.13 and 95% C.I. of 0.04 – 0.48 but authors interpreted it as ‘negatively associated with abscess formation’! The correct interpretation should have been ‘usage of insulin pen suggested a statistically significant evidence (at 5% level) of protection against abscess formation’.

- There are still many errors in interpretations vivid in the abstract section. Authors ought to correct them promptly!

2. A number of justifications for how data were handled and even analysed leave a lot to be desired.

e.g. authors justified age categorization of 0-24, 25-44 as well as ≥ 45 years to reflect distinct periods… (pp. 11 under sub-heading DATA ANALYSIS – lines 3-5 in the first paragraph). They also reported the categorization “to facilitate more interpretable analysis”. That is erroneous since:

a. Standard older age categorization that are acceptable even by the UN has a threshold of ≥ 65 years old and not 45 years used!

b. There is evidence that something is wrong with the usage of estimates of age on the outcome variable

e.g. didn’t the authors find it odd that there is a reversal of association between the

reported ‘cPR’ and ‘aPR’ as seen in table 6 in the results section?

3. Results section still display ambiguities in the estimates and their standard errors as justified by 95% C.I. values.

e.g. some variables have their ‘aPR’ greater than ‘cPR’, something that suggest a statistical indicator of ‘oddity’ on a logical sense.

Possible reasons: OVER-DISPERSION phenomena and/or MODEL MISSPECIFICATIONS due to potential but SIGNIFICANT confounding in the fitted model!

4. Reference section needs additional checks for they do not conform to Vancouver referencing style!

e.g. It was not clear whether authors used a REFERENCE MANAGER (or maybe assistance from an AI?) since the reference is not legible in standard English. Likewise, for references 14, 24 etc.

**Do you want your identity to be public for this peer review?** For information about this choice, including consent withdrawal, please see our Privacy Policy

Reviewer #5: **Yes:** Kelvin Melkizedeck Leshabari

---

## [Editor Report · Acceptance letter]

PONE-D-25-08926R3

PLOS One

Dear Dr. Mbwambo,

I'm pleased to inform you that your manuscript has been deemed suitable for publication in PLOS One. Congratulations! Your manuscript is now being handed over to our production team.

Kind regards,

on behalf of

Dr. Mohammed Misbah Ul Haq

Academic Editor

PLOS One